# Veterinary Clinics as Reservoirs for *Pseudomonas aeruginosa*: A Neglected Pathway in One Health Surveillance

**DOI:** 10.3390/antibiotics14070720

**Published:** 2025-07-17

**Authors:** George Cosmin Nadăş, Alice Mathilde Manchon, Cosmina Maria Bouari, Nicodim Iosif Fiț

**Affiliations:** Department of Microbiology, Immunology and Epidemiology, Faculty of Veterinary Medicine, University of Agricultural Sciences and Veterinary Medicine, 400372 Cluj-Napoca, Romania; gnadas@usamvcluj.ro (G.C.N.); alice.manchon76@gmail.com (A.M.M.); nfit@usamvcluj.ro (N.I.F.)

**Keywords:** *Pseudomonas aeruginosa*, veterinary infection control, biofilm resistance, nosocomial pathogens, One Health surveillance, environmental reservoirs

## Abstract

*Pseudomonas aeruginosa* is a highly adaptable opportunistic pathogen with significant clinical relevance in both human and veterinary medicine. Despite its well-documented role in hospital-acquired infections in human healthcare settings, its persistence and transmission within veterinary clinics remain underexplored. This review highlights the overlooked status of veterinary facilities as environmental reservoirs and amplification points for multidrug-resistant (MDR) *P. aeruginosa*, emphasizing their relevance to One Health surveillance. We examine the bacterium’s environmental survival strategies, including biofilm formation, resistance to disinfectants, and tolerance to nutrient-poor conditions that facilitate the long-term colonization of moist surfaces, drains, medical equipment, and plumbing systems. Common transmission vectors are identified, including asymptomatic animal carriers, contaminated instruments, and the hands of veterinary staff. The review synthesizes current data on antimicrobial resistance in environmental isolates, revealing frequent expression of efflux pumps and mobile resistance genes, and documents the potential for zoonotic transmission to staff and pet owners. Key gaps in environmental monitoring, infection control protocols, and genomic surveillance are identified, with a call for standardized approaches tailored to the veterinary context. Control strategies, including mechanical biofilm disruption, disinfectant cycling, effluent monitoring, and staff hygiene training, are evaluated for feasibility and impact. The article concludes with a One Health framework outlining cross-species and environmental transmission pathways. It advocates for harmonized surveillance, infrastructure improvements, and intersectoral collaboration to reduce the risk posed by MDR *P. aeruginosa* within veterinary clinical environments and beyond. By addressing these blind spots, veterinary facilities can become proactive partners in antimicrobial stewardship and global resistance mitigation.

## 1. Introduction

*Pseudomonas aeruginosa (P. aeruginosa)* is a metabolically versatile, Gram-negative bacillus that poses significant clinical challenges due to its intrinsic resistance mechanisms, adaptability to diverse environments, and capacity to cause opportunistic infections in both humans and animals [1,2,3]. In veterinary medicine, it is frequently isolated from cases of otitis externa, chronic wounds, urinary tract infections, and postoperative complications, particularly in small animal practice. These infections are often persistent and difficult to treat, requiring prolonged antimicrobial therapy and, in some cases, surgical intervention. Outcomes are exacerbated by the organism’s biofilm-forming ability and antimicrobial resistance (AMR) [4,5].

In the veterinary clinical environment, *P. aeruginosa* can act as a nosocomial pathogen, surviving on surfaces, instruments, and water sources. Unlike typical transient contaminants, it can adhere to abiotic surfaces and develop biofilms—complex, surface-attached microbial communities surrounded by an extracellular matrix composed of polysaccharides, proteins, and DNA. Within this matrix, the bacteria exhibit increased tolerance to antibiotics and disinfectants, allowing them to survive cleaning protocols that would typically eliminate planktonic cells [6]. Biofilm-associated *P. aeruginosa* also displays altered gene expression profiles, upregulation of efflux pumps, and downregulation of porins, contributing to its multidrug-resistant phenotype [7].

From a clinical standpoint, *P. aeruginosa* infections in animals are often secondary or opportunistic, developing in wounds with delayed healing, surgical sites, or in cases where the host’s defenses are compromised by chronic diseases or immunosuppressive therapy [4,8]. In these contexts, infection may manifest with characteristic green-blue pus, foul odor, necrotic tissue, and poor response to standard empirical antibiotic regimens. Delayed diagnosis and inappropriate initial therapy can lead to treatment failure, prolonged recovery, or even euthanasia in severe cases [9].

Despite these concerns, relatively few studies have focused on the veterinary hospital environment as a reservoir for *P. aeruginosa*. This contrasts with human hospitals, where extensive research has documented the organism’s ability to colonize sinks, drains, mechanical ventilators, and reusable medical equipment. Veterinary clinics, by comparison, often lack formal infection prevention and control programs, and environmental sampling is rarely performed outside of research settings. The lack of routine environmental monitoring may allow persistent environmental strains of *P. aeruginosa* to evade detection, increasing the risk of patient-to-patient or environment-to-patient transmission [10,11].

Clinically, the presence of *P. aeruginosa* in the veterinary environment raises critical concerns. Surgical site infections and wound dehiscence linked to contaminated instruments or hospital surfaces have been documented, though often only in the form of isolated case reports. There is growing recognition of the need for standardized cleaning protocols, staff training in infection control, and stewardship of antimicrobial use, yet these remain inconsistently applied across veterinary settings [12,13].

From a One Health perspective, veterinary clinics are potential hotspots for the amplification and dissemination of resistant strains. The shared spaces between pets, veterinary staff, and owners, especially in high-traffic urban clinics, create numerous interfaces where *P. aeruginosa* may spread. Human infections caused by animal-associated strains remain poorly documented; however, molecular studies have identified clonal relationships and shared resistance genes, suggesting that zoonotic transfer, although rare, is biologically plausible [14,15].

Recent surveillance shows that *P. aeruginosa* is far from a niche problem in veterinary medicine. Data indicate that the genus accounts for ≈ 10% of all bacterial isolates from dogs and 8% from horses submitted to diagnostic laboratories in 2021 [16]. In small-animal practice, its clinical footprint is even sharper in specific syndromes: culture-positive canine otitis externa yields *P. aeruginosa* in 25–41% of cases [9], while retrospective multicenter data suggest the organism causes ≈ 8% of documented infections in routine canine submissions. For urinary-tract infections, pooled estimates place their share between 3% and 8% of isolates worldwide [17]. These figures highlight the clinical relevance of *P. aeruginosa* across companion-animal species and justify the focused attention given in the pages that follow.

This review seeks to bridge the gap in understanding the environmental persistence and clinical significance of *P. aeruginosa* in veterinary settings. We examine the organism’s microbiological survival strategies, including biofilm formation and resistance mechanisms, alongside the clinical impact of infections it causes in companion animals. Additionally, we highlight infection control challenges, gaps in environmental surveillance, and potential public health implications. Through a multidisciplinary lens, we argue that veterinary hospitals must be recognized as critical control points for the management of *P. aeruginosa*, with far-reaching relevance for animal and human health alike.

## 2. Materials and Methods

### 2.1. Literature Search

A systematic search was performed in PubMed, Web of Science, and Scopus to capture primary research on *P. aeruginosa* epidemiology, pathogenesis, antimicrobial resistance, and therapeutic approaches. The coverage spanned 1 January 2000–31 March 2025, and the search was last updated on 1 April 2025. Search strings combined controlled vocabulary (MeSH) with free-text terms for *P. aeruginosa* linked by Boolean operators to thematic keywords (“pathogenesis”, “virulence”, “resistance”, “therapy”, “epidemiology”).

### 2.2. Eligibility Criteria

Records were eligible if they (i) presented original data on *P. aeruginosa* in humans, animals, or in vitro models and (ii) addressed at least one of the thematic domains listed above. All search results were exported to EndNote v20 and de-duplicated automatically, then verified manually.

### 2.3. Data Extraction

A piloted spreadsheet captured study design, setting, sample source, strain typing method, resistance profile, key outcomes, and reported limitations. One reviewer extracted all data; a second reviewer independently checked 10% of entries at random, with an error rate <2%. Any discrepancies were corrected by consensus.

### 2.4. Quality Appraisal

Because the corpus encompassed both experimental and observational designs, methodological quality was assessed with design-specific tools: the revised QUADAS-2 checklist for diagnostic studies and the Newcastle–Ottawa Scale for cohort and case–control studies. Each domain was rated “low”, “unclear”, or “high” risk of bias; disagreements were resolved by consensus.

### 2.5. Data Synthesis

Given the heterogeneity of study designs and outcome measures, results were synthesized narratively. Where at least three studies reported a comparable quantitative outcome (e.g., prevalence of carbapenem resistance), proportional data were pooled using a random-effects model (DerSimonian-Laird).

### 2.6. Sensitivity and Subgroup Analyses

Pre-planned subgroup analyses examined differences by (i) host species (human vs. animal), (ii) geographical region (continent-level), and (iii) clinical versus environmental sampling sites.

## 3. Environmental Survival and Adaptation Mechanisms

*P. aeruginosa* is uniquely equipped to persist in clinical environments, particularly those characterized by high humidity, repeated human–animal contact, and limited disinfection control—conditions commonly found in veterinary hospitals [16]. Its ecological success stems from a combination of intrinsic and adaptive features that enable it to colonize a broad range of abiotic surfaces, survive nutrient-limited conditions, and resist a variety of physical and chemical stressors [7,10].

One of *P. aeruginosa*’s best-studied survival tactics is biofilm formation, a structured, surface-attached community encased in a self-produced extracellular polymeric substance (EPS). Biofilms let the bacterium cling to stainless steel, plastic, glass, and other surfaces common in veterinary clinics (examination tables, surgical instruments, water bowls). Once established, they withstand disinfectants, desiccation, shear forces, and many antibiotics [18]. When high-touch tools such as mechanical ventilators, respirators, catheters, endoscopes, drains, and infusion pumps become contaminated, they serve as long-lived reservoirs for hospital-adapted *P. aeruginosa* clones, persisting even after routine cleaning [19,20,21].

In contrast to many veterinary pathogens that are susceptible to drying or require specific host-related conditions, *P. aeruginosa* has minimal nutritional requirements and thrives in both moist and nutrient-poor microenvironments. It can survive in distilled water for months and in saline solutions, catheter flushes, and even surface residues from improperly dried instruments. These traits elevate its status as a high-risk environmental contaminant, especially in multi-animal housing units and treatment areas with high patient turnover [10,22,23].

Table 1 summarizes the environmental survival properties and zoonotic potential of five bacterial species commonly encountered in veterinary settings. *P. aeruginosa* is noted for its exceptional ability to persist on moist surfaces and form robust biofilms that confer resistance to disinfectants, making it a significant nosocomial threat in veterinary hospitals [24]. *Staphylococcus pseudintermedius*, a coagulase-positive staphylococcus commonly associated with canine skin infections, has moderate biofilm capacity and environmental stability, with rare but documented zoonotic potential [25]. *Acinetobacter baumannii*, although primarily a concern in human healthcare, has emerged as a multidrug-resistant contaminant in veterinary ICUs, capable of prolonged survival in dry and moist environments [26]. *Enterococcus faecalis* and *Escherichia coli*, both opportunistic pathogens of the gastrointestinal tract, are capable of surviving on dry surfaces for extended periods and are increasingly recognized for their ability to exchange antimicrobial resistance genes across species barriers [27,28].

Medical respirators and their ventilator circuits provide a persistently humid micro-environment in which *P. aeruginosa* rapidly establishes and flourishes. Within the first 48 h of mechanical ventilation, the bacterium forms multilayered biofilms on endotracheal-tube plastics, embedding cells deep inside an extracellular polymeric matrix that ordinary re-processing cycles fail to penetrate [29]. Heated humidifiers further nurture these communities by delivering a continuous, warm moisture film that sustains growth even when only sterile saline or condensate is present. Viable *P. aeruginosa* can remain detectable in inspiratory and expiratory limbs, Y-adaptors, and humidifier chambers for at least a week after standard cleaning, highlighting the organism’s minimal nutritional requirements [30]. During coughing, suctioning, or circuit disconnection, aerosolized biofilm fragments and reverse airflow can propel cells upstream, seeding successive patients and elevating the risk of ventilator-associated pneumonia. Silver-impregnated circuits and antimicrobial coatings reduce but do not abolish colonization, highlighting the resilience of the biofilm mode of growth [31]. Veterinary ventilators, often refurbished from human ICUs and reused across multiple animal species, become persistent cross-animal infection hubs if single-use circuits or high-temperature sterilization protocols are not enforced [32]. Regular surveillance cultures, meticulous drying of circuits between cases, and periodic replacement of humidifier chambers are essential to break this biofilm-driven contamination cycle.

In short, *P. aeruginosa* is the environmental “survivor-in-chief” of veterinary hospitals. It (i) outlasts most rivals on damp or even nutrient-free surfaces for weeks to months, (ii) rapidly builds biofilms that shrug off routine disinfectants, and (iii) thrives in equipment condensate, especially ventilator circuits, where minimal nutrients suffice. These traits, combined with documented cross-patient spread, explain why *P. aeruginosa* dominates the persistence column in Table 1 and poses a disproportionate nosocomial threat when infection-control lapses occur.

## 4. Sources and Transmission in Veterinary Settings

*P. aeruginosa* circulates through veterinary clinics by exploiting a self-reinforcing triad of (1) shedding patients, (2) contaminated surfaces, fluids, and equipment, and (3) the hands and tools of staff. Understanding how these three legs interact is critical for breaking the cycle.

Dogs, cats, horses, birds, and exotic mammals can all carry *P. aeruginosa* asymptomatically in the oropharynx, conjunctiva, gut, or skin folds, or they may present with overt otitis, dermatitis, urinary tract infections, or wound infections. Every cough, sneeze, drip of otic exudate, or splash of urine deposits the bacterium onto cage bars, litter, bedding, and handler gloves. Because the organism multiplies in droplets and damp fur, even a lightly colonized pet can release thousands of cells during routine grooming, nebulization, or physiotherapy sessions. Shedding intensifies under antibiotic pressure that suppresses competing flora, making ICU patients on broad-spectrum therapy especially potent reservoirs [33,34,35].

Once on an abiotic surface, *P. aeruginosa* quickly adheres and, within hours, initiates a biofilm, particularly in the micro-scratches of stainless-steel tables, the silicone walls of water bowls, or the hydrophobic plastics of IV pump buttons. Moisture trapped under absorbent pads or in chewing-toy crevices nurtures exponential growth without visible slime, so staff may underestimate contamination. New patients placed on inadequately cleaned mats or plugged into the same ventilator circuit acquire organisms directly to wounds, endotracheal tubes, conjunctiva, or paws. Water systems amplify the problem: splash zones around sinks, dental units, hydrotherapy tanks, and mop buckets disseminate cells via aerosols and floor runoff into adjacent kennels [32,36].

Veterinarians, technicians, kennel attendants, and student observers bridge every ward and procedure room dozens of times per shift. Gloved hands that have handled a moist otitis case may carry residual *P. aeruginosa* through glove pinholes or cuff edges; bare-handed phone use immediately after pet restraint transfers organisms to touch screens and keyboards. Clippers, grooming combs, otoscope cones, rectal thermometers, stethoscope diaphragms, and handheld ultrasound probes often receive only cursory wiping between animals, creating high-throughput fomites. Even single-use items, such as syringes for catheter flushes and nebulizer masks, can become indirect vectors if placed on contaminated countertops before they reach the next patient [37,38].

High-density boarding areas, emergency wards with rapid patient turnover, and shared procedure rooms compress cleaning windows and promote shortcut disinfection practices. Central oxygen and vacuum lines, wall humidifiers, and floor drains form interconnected, moisture-rich niches that allow *P. aeruginosa* to migrate unseen between wings. Portable equipment carts, laundry hampers, and staff uniforms pick up organisms from one zone and seed them in another [39,40,41].

Effective control demands simultaneous action on all three fronts: (a) early identification and isolation of shedding patients; (b) meticulous, moisture-focused surface decontamination and adequate drying time between occupants; and (c) rigorous hand hygiene plus single-patient or terminally sterilized tools wherever feasible. Environmental microbiological surveillance, sink swabs, cage-door imprint cultures, and glove prints help locate silent reservoirs, while staff training and workflow redesign ensure that each leg of the triad is consistently disrupted [42,43].

## 5. Studies and Surveillance Gaps

Over the past two decades, the literature on *P. aeruginosa* in veterinary hospitals has grown sporadically yet unevenly. Most studies have been single-center, cross-sectional culture surveys that focused on one ward or a small cluster of high-risk surfaces, such as sink drains, treatment tables, and otoscope cones. Sample sizes have ranged from a dozen to a few hundred swabs, with follow-up periods rarely exceeding three months. Even within this narrow frame, methodologies vary: aerobic plate counts on cetrimide agar dominate older work, while more recent reports employ chromogenic media or qPCR to shorten turnaround times and recover viable but non-culturable cells [10,11,44].

The cumulative picture that emerges is fragmented but telling. Positivity rates for *P. aeruginosa* differ by two orders of magnitude: <1% on dry countertops in routine wards and up to 80% in moisture-rich drains of intensive care units. Multidrug resistance appears more frequently in isolates from surgical theaters and dentistry suites than in general wards, suggesting selective pressure from topical antimicrobials and disinfectants. Yet, because most studies capture only a single snapshot in time, it is impossible to determine whether these hot spots are persistent reservoirs or transient blooms linked to temporary workflow lapses. Longitudinal investigations remain scarce, and where they exist, they are limited to pilot projects that run for six to nine months, insufficient to reveal seasonal patterns or the impact of facility renovations [10,40,45].

The veterinary sector currently lacks consensus on what to sample, how often, and by which analytic method. Choices of swab type, pre-moistening solution, and sampling area are left to local preference, making meta-analysis almost impossible. Contact plates, sponge sticks, and liquid rinsates are used interchangeably, although each captures a different microbial fraction. Frequency likewise fluctuates dramatically: some hospitals perform quarterly point-prevalence screens; others sample only in response to confirmed clinical cases. Without agreed-upon minimum sampling frames, number of sites per square meter, mandatory inclusion of plumbing fixtures, or pre- and post-disinfection comparisons, data sets cannot be combined to produce benchmark rates or risk maps [46,47]. Analytical endpoints further complicate interpretation. Colony counts expressed as CFU/25 cm^2^ cannot be directly compared to qPCR cycle-threshold values or next-generation sequencing read depths. Antimicrobial susceptibility panels differ as well: some laboratories test only trimethoprim-sulfonamides and fluoroquinolones, while others include extended-spectrum beta-lactams and polymyxins. This heterogeneity hinders the early recognition of emerging resistance clusters and undercuts the development of evidence-based cleaning policies [48,49,50].

Human hospitals follow clear, detailed rules for checking their surroundings for germs. The guidelines spell out when to collect samples and where to swab, both frequently touched items such as bed rails and infusion pumps, and wet, high-flow spots like sink drains and ice machines. They also list which bacteria count as “alert organisms” and the levels that trigger action. Large centers back up this routine swabbing with rapid DNA typing, live data dashboards, and dedicated infection-control teams. By contrast, most veterinary clinics lack such resources: staff take environmental samples only when time allows or send them to outside labs on an ad hoc basis, and results can take weeks to return [51,52].

Another gap lies in feedback mechanisms. Human hospitals couple environmental data to mandatory remediation steps by repeat sampling after cleaning, temporary ward closures, or engineering fixes such as drain replacements. Veterinary settings may flag a positive site but lack a formal pathway for escalation, relying on informal communication chains that risk delay and incomplete follow-through. Hand-hygiene surveillance in human wards employs direct observation, electronic dispenser counters, and ultraviolet tracers; veterinary hospitals rarely deploy these tools beyond occasional audits [51,53,54].

Borrowing elements from the human model, such as risk-stratified sampling schedules, unified interpretive breakpoints, and rapid sequencing to link environmental and clinical isolates would accelerate progress. Yet direct transplantation is not trivial. Species diversity, patient behaviors, and the architectural differences between kennels and human rooms make one-to-one protocol mapping impractical. Customized standards must account for factors unique to animal care: shared water bowls, species-specific grooming tools, and the routine use of damp mopping rather than disposable floor pads [55,56,57].

In sum, the evidence base for *P. aeruginosa* environmental surveillance in veterinary medicine is thin, inconsistent, and often incomparable across studies. Establishing harmonized sampling guidelines, shared data repositories, and minimum analytic panels is an urgent prerequisite for understanding true prevalence, identifying persistent reservoirs, and measuring the impact of control interventions.

## 6. Antimicrobial Resistance in Environmental Isolates

Environmental *P. aeruginosa* isolates collected from kennel floors, drain biofilms, stainless-steel worktops, and handheld equipment routinely exhibit narrower phenotypic resistance panels than their clinical counterparts—yet they are far from benign. Surface strains tend to express high-level efflux pump activity and inducible chromosomal β-lactamases that confer low-to-moderate resistance to first-line agents such as aminopenicillins and early-generation cephalosporins. In contrast, wound or respiratory isolates often accumulate mobile genetic elements such as class 1 integrons, Tn7-like transposons, and carbapenemase plasmids, which extend resistance to carbapenems, fluoroquinolones, and polymyxins [10,11,58,59].

This divergence reflects the differing selective landscapes. Environmental populations experience intermittent, sub-inhibitory exposures to quaternary ammonium compounds, peroxide blends, and topical antibiotics shed in urine, favoring mutations in membrane permeability and redox–stress pathways. Clinical populations face sustained therapeutic drug concentrations that drive the acquisition of mobile resistance determinants. Nevertheless, genomic linkage studies repeatedly demonstrate a porous boundary between the two domains: environmental clones can seed initial colonization, while patient isolates returned to cages on fur or exudate reseed drains, creating a bidirectional resistance conveyor [7,60].

A binary heatmap tracks the weekly detection of a single, genotypically confirmed multidrug-resistant (MDR) *P. aeruginosa* clone in 10 fixed locations over a 12-week surveillance window. Rows represent individual sampling sites—five ICU sink drains (SINK-ICU-01 to -05) and five operating-room floor zones (FLOOR-OR-01 to -05). Columns correspond to consecutive calendar weeks (week 1 to week 12). Yellow cells denote culture-positive recovery of the MDR clone; purple cells indicate no detection.

Figure 1 reveals (i) sustained colonization of specific wet niches, e.g., SINK-ICU-02 and SINK-ICU-05, where the clone was recovered in ≥6 contiguous weeks; (ii) episodic spill-over onto operating-room floors during weeks 7–10, suggesting periodic environmental dissemination likely driven by workflow or cleaning lapses; and (iii) successful clearance of several sites (e.g., SINK-ICU-01) after week 6, consistent with targeted remediation. This temporal, site-resolved view highlights hidden reservoirs and helps infection-control teams time and target intensified decontamination or engineering fixes (e.g., drain replacement) before sporadic positives evolve into ward-wide outbreaks (see Appendix A).

Multidrug-resistant (MDR) clones gain a stealth advantage in veterinary hospitals because routine surveillance often targets symptomatic animals rather than environmental reservoirs. Within the oligotrophic tapwater film, MDR *P. aeruginosa* cells downshift into a persister state, displaying minimal metabolic activity and heightened tolerance to disinfectants. Standard aerobic plating may yield “negative” results even when thousands of viable but non-culturable cells inhabit microcrevices. Molecular assays, meanwhile, are rarely applied outside outbreak investigations, leaving long latency periods during which MDR lineages can smolder [48,61].

Persistence is further prolonged by the architectural features of veterinary plumbing. Horizontal drain runs and low flow rates allow for detritus accumulation, providing carbon sources and shelter from shear forces. Biofilm residence times measured by fluorescent-tracer studies exceed 60 days in some facilities, easily outlasting weekly cleaning cycles. In this context, sporadic positive cultures from patient wounds may be interpreted as isolated events when they are, in fact, flare-ups driven by an entrenched environmental lineage [40,62].

Effluent monitoring extends surveillance beyond the hospital’s walls and serves three strategic goals. First, it functions as an integrative sample, capturing shed organisms from sinks, floor drains, and washing machines in a single composite stream. Detecting carbapenemase genes or high-risk sequence types in wastewater can act as an early-warning flag that resistance is building upstream, even when point-source swabbing remains negative [63,64].

Second, effluent analysis addresses public health externalities. Veterinary hospitals discharge into municipal sewers that ultimately feed surface waters or agricultural irrigation systems. Studies of urban wastewater have traced veterinary-linked *P. aeruginosa* clones into downstream river sediments, where they mingle with human-derived strains, facilitating inter-species gene exchange. Regular testing, therefore, supports One Health stewardship by quantifying the hospital’s ecological footprint and guiding engineering controls such as on-site ozone or UV disinfection units [65,66].

Third, longitudinal effluent data provide a performance metric for infection-control interventions. A step-wise decline in resistance-gene copy number or viable pseudomonad counts following drain replacement, biocide rotation, or intensified hand-hygiene campaigns offers objective evidence that resource-intensive measures are paying dividends. Conversely, a rebound signals the need for root-cause analysis and program recalibration [67,68].

Together, granular surface sampling, the vigilant detection of silent MDR reservoirs, and effluent monitoring form a triad of defenses that closes the feedback loop between hospital practice and environmental impact, protecting animal patients, clinical staff, and the wider community alike.

## 7. Implications for Nosocomial Infections in Pets

Hospital-acquired *P. infections* in companion animals most commonly present as postoperative incisional cellulitis, chronic otitis externa/media, and delayed-healing traumatic wounds. In surgical patients, even a brief contact between a freshly sutured incision and a contaminated recovery mat can seed the dermis with biofilm-competent cells that remain quiescent until the first dressing change. Ototopical medications compounded in-house are another frequent conduit: once an ointment jar is opened in a treatment area that harbors aerosolized *P. aeruginosa*, residual droplets can colonize the base and subsequently be massaged deep into the external canal during dosing [69,70]. Topical antimicrobials or glucocorticoids selected for their broad-spectrum or anti-inflammatory effects paradoxically lower local immunity, permitting the bacteria to proliferate within hours and extend into the middle ear through a compromised tympanic membrane. Similar dynamics occur in bite wounds and pressure sores, where fibrinous exudate and necrotic debris offer a nutrient-rich scaffold for rapid biofilm maturation. Clinically, these infections manifest as malodorous exudation, green-tinged discharge, or ‘blue-paw’ staining of dressing materials, subtle clues that can be missed if clinicians default to empirical first-line antibiotics without culture confirmation [71].

Intensive care units (ICUs), oncology wards, and dermatology suites amplify the risk of nosocomial *P. aeruginosa* transmission because they combine three vulnerability drivers: (i) a high prevalence of patients with indwelling devices; (ii) continuous manipulation of respiratory and urinary tracts; and (iii) heavy use of broad-spectrum antimicrobials that suppress competing flora. Endotracheal tubes, central venous catheters, and urinary Foley lines act as both physical bridges and nutrient wicks, allowing organisms from sinks or humidifier water to ascend into normally sterile sites. Condensate builds up inside ventilator circuits; if staff empty the water traps without first changing gloves, they can inadvertently carry the pathogen from one infected patient to other ventilated patients nearby [72,73]. Oncology patients receiving cytotoxic drugs suffer neutropenia and mucosal barrier breakdown, making even transient bloodstream seeding clinically significant. Dermatology service areas, meanwhile, process large numbers of chronic otitis and pyoderma cases; the inevitable aerosolization during ear flushing or clip-and-scrub procedures distributes *P. aeruginosa* across benches, keyboards, and ophthalmoscope heads, creating a rolling chain of subclinical colonization events that feed back into the ICU when those same devices are deployed on critical patients [74].

Although formal outbreak reports remain relatively scarce compared with the human literature, the available case studies highlight the speed and scale at which *P. aeruginosa* can spread under the radar. One multicenter investigation linked a cluster of canine postoperative endophthalmitis cases to a single batch of non-sterile ophthalmic irrigation solution that had been aliquoted into multiple syringes on an unprotected prep counter [74]. Whole-genome sequencing later confirmed a near-identical strain in the unused portion of the bottle, the counter drain, and the affected eyes. Another episode in a referral ICU involved dogs developing ventilator-associated pneumonia within nine days; environmental sampling traced the source to a cracked plastic elbow in the shared humidifier line, and the outbreak ceased only after a complete circuit replacement and a temporary shift to heat–moisture exchanger filters. Smaller events, often unrecognized as outbreaks, include persistent otitis in grooming wards and cyclical contamination of hydrotherapy pools used for orthopedic rehabilitation. These examples collectively illustrate two principles: first, that environmental reservoirs frequently act as the outbreak nucleus; and second, that successful resolution hinges on engineering interventions (component replacement, plumbing redesign) as much as on pharmacological therapy [62,73,75].

In Figure 2, by combining a surgical-site schematic (Panel A), an ICU device-transmission diagram (Panel B), and concrete outbreak timelines (Panel C), this figure provides a comprehensive visual narrative of how *P. aeruginosa* moves from environmental niches into veterinary patients. Panel A focuses on contamination pathways in the OR, Panel B on high-risk device interactions in the ICU, and Panel C on documented outbreaks that reinforce the critical role of environmental reservoirs. Viewed together, these panels illustrate the multifactorial nature of nosocomial transmission and underscore the need for vigilant environmental monitoring, strict device protocols, and rapid engineering or procedural interventions to prevent and control infections in veterinary settings (see Appendix A).

## 8. Infection Control and Sanitation Practices

Effective disinfection is the cornerstone of interrupting *P. aeruginosa* transmission chains in veterinary facilities. However, not all disinfectants perform equally against this hardy Gram-negative organism. Quaternary ammonium compounds (QACs), widely used for routine surface cleaning, exhibit only moderate activity and may select for tolerant strains when applied at suboptimal concentrations or contact times [4]. Chlorine-based disinfectants (sodium hypochlorite) remain a reliable first-line choice due to their broad-spectrum bactericidal action; however, organic soiling can rapidly neutralize hypochlorite’s activity, making thorough pre-cleaning essential. Peroxygen agents (hydrogen peroxide and peracetic acid formulations) offer strong activity against planktonic *P. aeruginosa* and, in some concentrations, partial biofilm penetration [76]. Accelerated hydrogen peroxide solutions combine a low-foaming profile with rapid kill kinetics but require manufacturers’ prescribed dwell times (often 5–10 min) to achieve a >5-log reduction. Phenolic disinfectants maintain good activity in the presence of organic load but can be corrosive to certain surfaces and are less popular in veterinary settings due to residue concerns near animals [77,78].

Fogging or vaporized hydrogen peroxide (VHP) can achieve high-level disinfection of enclosed rooms but involves specialized equipment, room sealing, and staff training. VHP effectively eradicates *P. aeruginosa* from both planktonic and surface-attached states, including early biofilms, yet may be cost-prohibitive for many clinics [79]. Similarly, ultraviolet-C (UV-C) irradiation devices can serve as adjuncts to manual cleaning by inactivating residual vegetative bacteria on unobstructed surfaces; however, shadowed niches and irregular equipment geometry limit UV-C penetration. Selecting an appropriate disinfectant regimen, therefore, requires consideration of the following: (1) expected organic load and soiling; (2) surface material compatibility; (3) required contact time and dwell conditions; and (4) staff capacity for adherence to protocols. Daily rotation (“biocide cycling”) between QACs and peroxygen agents can mitigate selective pressure, reduce the risk of tolerance development, and improve overall kill rates [80,81].

*P. aeruginosa* biofilms pose a particular sanitation challenge because the extracellular polymeric matrix impedes the penetration of disinfectants and shelters persister cells. Mechanical disruption is often the most effective initial step: scrubbing with firm bristle brushes or abrasive pads under running water can dislodge mature biofilms from drain grates, floor joints, and textured countertops. Enclosing drains, traps, and sink barrels within removable sleeves or sleeves with smooth internal linings reduces surface roughness, making periodic scraping more efficient. For in situ biofilm control, enzymatic detergents containing proteases, lipases, and DNases can degrade the biofilm matrix, rendering embedded cells more susceptible to subsequent disinfectant application. These enzymatic products should be applied to pre-cleaned surfaces, allowed to dwell according to manufacturer guidelines, and then rinsed thoroughly before applying a chemical disinfectant [82,83,84].

Surfactant-based cleaners containing nonionic or amphoteric surfactants serve as valuable adjuncts by lowering surface tension and facilitating the removal of residual organic debris, but they should never replace a true bactericidal product. For heavily colonized plumbing fixtures, periodic “shock” treatments involving high concentrations of peracetic acid or chlorine dioxide can flush drain lines, though such approaches must be balanced against the potential corrosion of metal pipes [85]. In critical areas, like surgical suites and ICUs, dedicated “drain disinfection kits” that combine a small brush with a premeasured oxidizing foam can ensure consistent coverage while minimizing staff exposure to concentrated chemicals [86]. Finally, some facilities have experimented with continuous low-dose antimicrobial coatings (e.g., silver nanoparticles embedded in polymer films) on high-risk touchpoints (door handles, faucet knobs). While preliminary data suggest reduced biofilm attachment, routine monitoring and evaluation are needed to confirm long-term efficacy and to detect whether *P. aeruginosa* develops adaptive tolerance [87].

Recommendations for surface decontamination and hand hygiene.

A standardized, multi-step approach yields the most reliable outcomes. First, visible organic matter (hair, blood, exudate) should be removed using disposable towels or absorbent pads. Second, a detergent cleaning, with mechanical action (spray and wipe), removes residual soil and biofilm matrix. Third, the application of a broad-spectrum disinfectant is carried out according to the manufacturer-specified dilution, application method (spray, wipe, or foam), and dwell time. Fourth, surfaces are rinsed or wiped if required to remove toxic residues, especially on counters where patients may be examined. High-touch surfaces (faucet handles, doorknobs, light switches) demand twice-daily attention, whereas low-touch areas (walls above 1 m, ceilings) can follow a weekly schedule. Shared portable equipment (stethoscopes, ultrasonography probes, otoscope heads) should be cleaned and disinfected between each patient. For ultrasound probes, use probe covers for sterile procedures and disinfect with a compatible low-level disinfectant after every use; avoid submerging electrical connectors [39,75,88].

Hands are the most common vector for transferring *P. aeruginosa* between patients and fomites. Alcohol-based hand rubs (ABHRs) with at least 60% ethanol or isopropanol by volume are the preferred method when hands are not visibly soiled. ABHRs act rapidly, evaporate without residue, and achieve broad antimicrobial kill if applied correctly (20–30 s of friction). For visibly soiled hands or after contact with bodily fluids, hands must be washed with soap and water for at least 20 s; friction and running water physically remove bacteria and organic matter. Nail brushes should be employed under fingernails during surgical patient care. Fingernails should be kept short (< 0.5 cm), and artificial nails or extenders should be prohibited in high-risk areas. Jewelry (rings, bracelets) should be removed to facilitate thorough hand coverage [89,90].

To improve compliance, ABHR dispensers should be mounted at key points: room entrances/exits, just outside exam rooms, at the head of each procedure table, and near kennel entrances. Visual cues, brightly colored thumbprint decals on dispensers, and floor arrows guiding staff through the hand-hygiene sequence help reinforce correct technique. Electronic monitoring systems or “smart pumps” can log each dispense event; weekly compliance reports should be shared with staff to encourage peer accountability. Finally, hand-hygiene training should include periodic direct observation audits using a standardized checklist (e.g., WHO’s “5 Moments for hand hygiene” adapted for veterinary care). Immediate feedback and spot rewards (e.g., stickers on badges) sustain long-term behavior change and reduce the risk of *P. aeruginosa* transmission [91,92].

## 9. One Health Perspective and Zoonotic Considerations

Veterinary clinics occupy a critical intersection between animal, human, and environmental health domains. Patients arrive carrying diverse microbiota, including *P. aeruginosa* strains acquired at home or in previous facilities. Once inside the clinic, animals encounter high-density housing, shared equipment, and frequent human handling—conditions that amplify opportunities for resistant organisms to proliferate and disseminate. High-touch surfaces (kennel bars, exam tables) and wet sites (sink basins, bathing tubs) serve as convergence points where bacteria from multiple patients mix. Handlers, technicians, and veterinarians inadvertently shuttle microbes among these reservoirs, while shared tools (otoscopes, stethoscopes, thermometers) act as fomites [10,75,93,94].

In such an ecosystem, a single MDR *P. aeruginosa* isolate carried in one pet’s ear or wound can rapidly seed drains, countertops, and cages. From these foci, resistant clones may colonize subsequent patients, particularly if disinfectant protocols are suboptimal or biofilms have already become established. Furthermore, veterinary clinics often sublease or share water and sewer lines with adjacent human healthcare or municipal services; plumbing interconnections can facilitate the downstream migration of resistance genes into broader community sewer systems. Laundry operations that process soiled pet linens may also discharge viable bacteria into wastewater if not pretreated properly [75,93,94,95].

Taken together, these factors position veterinary facilities as amplification nodes where environmental, animal, and human bacterial populations intersect. In the absence of rigorous One Health-oriented surveillance, veterinarians may inadvertently contribute to the expansion of MDR *P. aeruginosa* reservoirs that affect both animal and human populations.

The zoonotic transmission of *P. aeruginosa* is often underappreciated because most human infections are attributed to patient-to-patient spread within hospitals. Nonetheless, contact with colonized or infected pets can expose veterinary staff and pet owners to the same clones circulating in the clinic environment. Technicians who handle ear flushes, wound dressings, or catheterized patients face particular risk, as glove microtears and contaminated sleeves can deposit bacteria onto skin or clothing. Aerosolization during bathing or hydrotherapy sessions can also suspend bacteria in small droplets, which may be inhaled or settle on mucous membranes [3,57,60,96].

For immunocompromised employees (e.g., those receiving chemotherapy or with uncontrolled diabetes), transient colonization can progress to invasive disease, ranging from otitis externa to pneumonia or septicemia. Similarly, owners who assist with home care (ear cleanings, wound bandage changes) can introduce clinic-acquired strains into the household. Pets living in the same residence may become secondary carriers, creating a feedback loop that perpetuates persistent colonization cycles [60,97].

Moreover, the veterinary hospital serves as a crossroads for multiple animal populations: owned pets, stray or shelter animals brought in for foster or treatment, and wildlife patients in some teaching facilities. Wildlife reservoirs—such as rodents or birds—can acquire clinic-adapted *P. aeruginosa* strains and spread them back into urban or peri-urban environments. Shelter-bound cats or dogs that pass through the clinic may carry resistant organisms to adoption sites or foster homes, further broadening the zoonotic and interspecies transmission landscape [10,75,98].

Although veterinary and human medical infrastructures often operate independently, they are linked through shared bacterial pools, overlapping staff networks, and sometimes shared supply chains for disinfectants or pharmaceuticals. Clinic technicians who work part-time in human hospitals or veterinary staff who inadvertently bring contaminated scrubs home may introduce MDR *P. aeruginosa* into the human healthcare setting. Conversely, human hospital workers who treat pets or volunteer at animal shelters can import resistant strains back into veterinary wards [41,57,93].

Pharmaceutical products present another bridge: compounded otic or topical preparations manufactured in one facility may be supplied to both human clinics (e.g., for veterinary technicians’ pets) and veterinary hospitals. If contamination occurs at the compounding stage, identical resistant strains can enter parallel animal and human treatment environments. Also, veterinary clinics that share veterinary teaching spaces with medical students or run combined research programs may have interwoven ventilation, drainage, or waste-handling systems. In such scenarios, a resistant clone established in a veterinary ICU could infiltrate a nearby human ICU via plumbing cross-connections or shared equipment storage rooms [93,95,99,100].

Finally, animal-derived food products (e.g., raw pet diets prepared on-site) can harbor *P. aeruginosa* that then enter human kitchens or dining areas when handled by staff. Without strict separation, these bacterial strains can contaminate food preparation zones, potentially leading to sporadic foodborne colonization in staff or family members [101,102].

Given these interdependencies, a One Health approach demands that veterinary facilities coordinate infection control efforts with local public health agencies, human hospitals, and environmental services. Sharing resistance surveillance data, standardizing molecular typing protocols, and jointly addressing wastewater treatment can disrupt the cross-sectoral flow of MDR *P. aeruginosa*. Clear communication channels between veterinary and human healthcare providers ensure that outbreak signals in one domain prompt swift investigation and intervention in the other, breaking the chain of transmission before resistant strains gain a stronger foothold.

This conceptual diagram (Figure 3) highlights the multifaceted transmission routes through which multidrug-resistant *P. aeruginosa* circulates between animals, humans, and the environment in a veterinary clinical context. It emphasizes the clinic’s role not merely as a point of care but as a central node within a larger One Health network where resistant strains can be acquired, amplified, and redistributed. The overlapping pathway, through contaminated hands, shared surfaces, animal discharge, and infrastructural links such as plumbing or pharmaceuticals, highlights that effective control requires coordinated action across disciplines. Visualizing these interactions reinforces the urgency of integrated surveillance systems and cross-sector communication to intercept MDR clones before they gain an ecological foothold across species boundaries (see Appendix A).

## 10. Research Gaps and Future Directions

### 10.1. Need for Environmental Surveillance Protocols

Despite the increasing recognition of *P. aeruginosa* as a persistent contaminant in veterinary environments, there is still no standardized protocol for routine environmental surveillance [93]. Current studies vary widely in sampling frequency, site selection, and analytical techniques, making cross-facility comparisons nearly impossible. Many veterinary hospitals rely on reactive testing, triggered by clinical cases or outbreak suspicion, rather than proactive baseline mapping [96]. Key high-risk reservoirs such as ventilators, respirators, floor drains, faucet aerators, and grooming tools are inconsistently sampled or excluded altogether [103]. To address this, consensus guidelines are urgently needed, defining minimum sampling areas per square meter, the required inclusion of wet zones, and thresholds for interpreting positivity. Without harmonized methods, data from different clinics cannot be integrated to track regional trends or benchmark infection-control performance.

### 10.2. Genomic Studies Comparing Environmental and Clinical Isolates

Genomic epidemiology has revolutionized outbreak investigations in human healthcare but remains underutilized in veterinary settings [94]. Most environmental *P. aeruginosa* isolates are still assessed solely via phenotypic antimicrobial susceptibility or basic PCR assays. Whole-genome sequencing (WGS) or core-genome MLST could reveal whether specific clones persist over time, jump between patients and surfaces, or acquire resistance determinants in situ [104]. Comparative genomics of clinical and environmental isolates from the same facility would clarify transmission pathways and detect recombination events, plasmid exchange, or biofilm-associated adaptations. The inclusion of long-read sequencing could further resolve structural elements like integrons or resistance islands [105]. To support broader implementation, shared repositories and cloud-based platforms for WGS data should be developed specifically for veterinary institutions, enabling the global mapping of high-risk *P. aeruginosa* lineages [106].

### 10.3. Standardizing Infection Control in Veterinary Settings

Veterinary hospitals lack the infection-prevention infrastructure common to human healthcare facilities [54]. Most operate without dedicated infection-control personnel, formal audit systems, or detailed cleaning standard operating procedures (SOPs) adapted to species-specific behavior and housing. Even within large referral centers, disinfection protocols often reflect manufacturer guidelines rather than organism-specific kill data, and hand-hygiene compliance is rarely monitored quantitatively [104]. Developing a core framework for infection control in veterinary environments would require input from clinical microbiologists, epidemiologists, and veterinary practitioners across small and large animal specialties [107]. Standardizing contact times, surface compatibility matrices, and tool sterilization workflows would support broader adherence and training. Incorporating One Health principles into this framework, such as effluent management, interspecies contact risk, and staff cross-contamination, will future-proof guidelines as zoonotic threats continue to evolve.

### 10.4. Limitations and Generalizability

Our synthesis has several constraints. First, many of the surveillance datasets discussed here originate from a single hospital or regional laboratory. Such single-site designs capture local strain dynamics, facility architecture, and antimicrobial usage patterns that may not mirror conditions elsewhere. Consequently, extrapolating their findings to other geographical regions, animal populations, or practice types should be done with caution, and confirmation in large, multicenter, longitudinal cohorts is warranted. Second, diagnostic and sampling methodologies varied markedly among studies—from swab types to breakpoint standards, introducing measurement inconsistency that limits direct comparability. Third, clinical metadata (e.g., patient comorbidities, prior antimicrobial exposure) were often incompletely reported, preventing nuanced risk-factor analyses. Fourth, most investigations provided cross-sectional snapshots rather than true longitudinal follow-up, precluding robust inference on temporal trends and causality. Finally, our search primarily focused on full-text articles published in English, so relevant data in other languages or the grey literature may have been overlooked, potentially introducing publication bias. Collectively, these constraints emphasize the need for harmonized, multicenter surveillance networks and comprehensive reporting standards to strengthen future investigations of *P. aeruginosa* epidemiology and resistance evolution.

### 10.5. Priority Research Agenda

Research priorities for tackling multidrug-resistant *P. aeruginosa* in veterinary hospitals:Standardize environmental surveillance protocols—agree on core wet-zone sites, sampling frequency, and analytic thresholds to enable cross-facility benchmarking.Apply whole-genome sequencing (WGS)—systematically link environmental and clinical isolates and populate shared veterinary AMR databases.Design and validate species-appropriate infection-control SOPs—include audited hand-hygiene and surface-disinfection workflows tailored to different animal wards.Evaluate biofilm-targeted disinfection and engineering interventions—test enzymatic detergents, drain sleeves, and oxidizing “shock” flushes in prospective, real-world trials.Integrate effluent/wastewater monitoring into One Health surveillance—track the off-site dissemination of high-risk clones and resistance genes.Quantify zoonotic spill-over risk—run longitudinal staff-and-owner carriage studies with SNP-level source attribution to inform risk-based personal protective equipment (PPE) and home-care guidance.

## 11. Conclusions

Multidrug-resistant *P. aeruginosa* represents a persistent and often underestimated threat in veterinary healthcare settings. Its exceptional adaptability to moist, nutrient-limited environments, coupled with intrinsic resistance mechanisms and biofilm formation, allows it to persist undetected on surfaces, in drains, and within shared equipment. This review shows that veterinary clinics are ecological crossroads where animal, human, and environmental microbiomes converge; without strong infection-control measures, they can become amplifiers of antimicrobial resistance.

Despite growing awareness, major gaps remain: environmental surveillance is inconsistent, genomic linkage between clinical and environmental strains is rarely pursued, and many clinics lack standard operating procedures for sanitation and hand hygiene. The veterinary sector must move toward harmonized, evidence-based infection prevention frameworks designed to address its unique spatial, species, and operational constraints. Integrating One Health principles, such as effluent monitoring, cross-sectoral communication, and zoonotic risk mitigation, will be essential to future-proof antimicrobial stewardship.

Addressing these challenges demands collaboration: between veterinary hospitals and public health agencies, between clinicians and microbiologists, and between animal and human care sectors. With the right investment in surveillance, training, and infrastructure, the veterinary profession can not only mitigate the risk of nosocomial *P. aeruginosa* outbreaks but also contribute meaningfully to global efforts in hindering antimicrobial resistance.

## Figures and Tables

**Figure 1 antibiotics-14-00720-f001:**
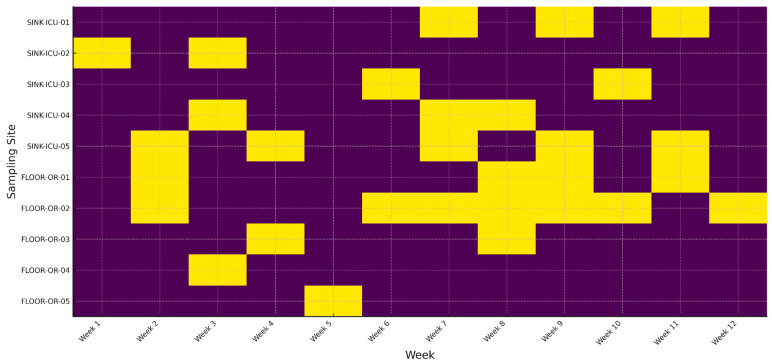
Long-term persistence of an MDR *P. aeruginosa* clone across hospital sites.

**Figure 2 antibiotics-14-00720-f002:**
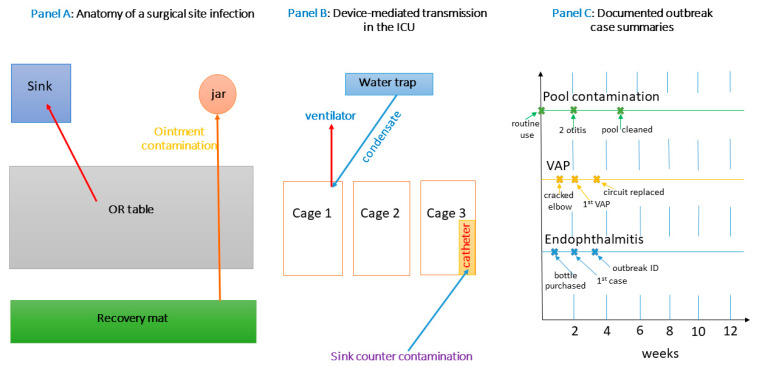
Multi-panel overview of nosocomial *P. aeruginosa* in veterinary settings.

**Figure 3 antibiotics-14-00720-f003:**
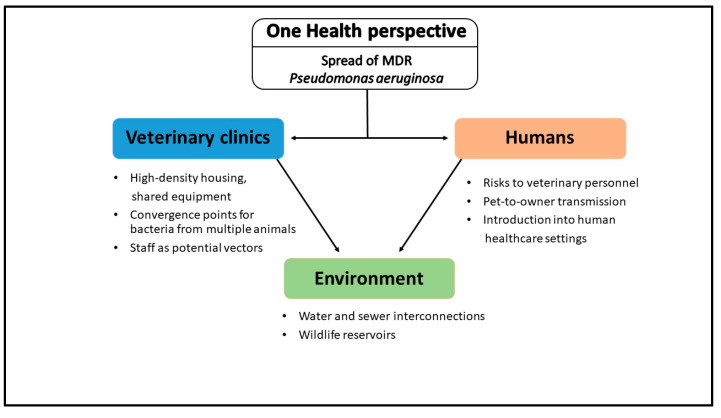
Conceptual diagram of One Health transmission pathways for MDR *P. aeruginosa*.

**Table 1 antibiotics-14-00720-t001:** Environmental persistence characteristics, biofilm capacity, disinfectant resistance, and zoonotic potential of selected veterinary-relevant bacterial pathogens.

Pathogen	Environmental Survival Time	Biofilm Formation	Resistance to Disinfectants	Veterinary Relevance	Zoonotic Potential
*Pseudomonas aeruginosa*	Up to months (moist surfaces)	Strong (dense EPS matrix)	High (especially in biofilms)	Wounds, otitis, and surgical site infections	Moderate to High
*Staphylococcus pseudintermedius*	Days to weeks	Moderate	Moderate	Canine pyoderma, abscesses, wounds	Low to Moderate
*Acinetobacter baumannii*	Weeks to months (dry/moist)	Strong	High	MDR hospital pathogen in animals	Moderate
*Enterococcus faecalis*	Weeks (especially dry surfaces)	Weak to Moderate	Variable	GI tract, wounds, urinary tract	Moderate
*Escherichia coli*	Days (especially moist)	Moderate	Moderate	UTIs, GI infections, and common contaminants	High

## Data Availability

Not applicable.

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
