# Peer review of "Veterinary Clinics as Reservoirs for Pseudomonas aeruginosa: A Neglected Pathway in One Health Surveillance"

_antibiotics, 2025, doi:10.3390/antibiotics14070720_

Round 1
Reviewer 1 Report
Comments and Suggestions for Authors
- Please briefly describe your literature search and selection methods to improve transparency and reproducibility.
- Acknowledge the limitations of generalizing findings from single-site surveillance studies.
- Consider summarizing research priorities in a bullet list or table to guide future work.
- Ensure all figures are present and properly captioned in the final version.
Author Response
Dear Reviewer 1,
Thank you very much for your thoughtful and insightful comments on our manuscript. We truly appreciate the time and expertise you invested in evaluating our work. Your constructive feedback has helped us identify several areas for improvement, and we have carefully addressed each point in the revised version. We believe these changes have strengthened both the clarity and the scientific rigor of the paper, and we are grateful for your contribution to this process.

Reviewer 2 Report
Comments and Suggestions for Authors
General description:
This manuscript presents a comprehensive and valuable review of Pseudomonas aeruginosa as an important nosocomial and environmental threat in veterinary clinics, with major relevance to One Health antimicrobial resistance surveillance. It brings attention to an area that hasn't been well studied, showing that veterinary clinics are often ignored but important places where resistant bacteria can grow and spread between animals, humans, and the environment.
General comments:
The topic is very important and fills a big gap in both veterinary and One Health research. The paper is clearly written, well-supported with up-to-date references, and brings together science, clinical experience, and policy ideas in a useful way. It clearly shows that veterinary clinics (state, private and faculties), are often major sources of resistant P. aeruginosa and makes a strong case for better monitoring and infection control. The idea of including original graphical design elements increases the quality of the manuscript. In addition, the supplementary files are appropriate and bring clarity in explanation of the figures.
Nonetheless, although of high quality in the scientific and description features, the visibility of picture 2 is average, with text that is dificult to read. The recommendation is to improve it. Moreover, some parts are a bit hard to read and use too much technical language, which might make it difficult for a wider audience to follow. Small changes in phrasing and structure would help make it clearer and easier to understand.
Specific comments:
Line 48: P. aeroginosa in not in italic, please change.
Line 75: the same as previous comment.
Lines 99-101: please add ventilators and respirators too. They are part of the clinical tools, but their importance is major, so they should be mentioned in the text.
Lines 200-205: please rephrase to make it easier to understand.
Line 240: Pseudomonas aeruginosa should be in italic
Lines 301-302: please rephrase
Line 479: Pseudomonas aeruginosa should be in italic
Lines 494-495: please add ventilators and respirators too
Lines 527-529: please rephrase.
Recommendation:
Minor revisions. The paper is solid, original, and important in the field. With a few changes to make it easier to read, improving the quality of picture 2, it will be a useful resource for both vets and researchers.
Author Response
Dear reviewer 2,
We would like to thank you sincerely for your thoughtful, constructive, and encouraging review. Figure 2 has been redesigned with larger fonts, higher contrast, and vector graphics to ensure legibility in both print and digital formats.
Your insights have helped us improve both the clarity and impact of the manuscript. Below we provide a point-by-point reply.

Reviewer 3 Report
Comments and Suggestions for Authors
This is a comprehensive and well-structured review that addresses a critical yet often underrecognized aspect of Pseudomonas aeruginosa transmission within veterinary clinical settings. The manuscript provides a thorough examination of the environmental persistence, resistance mechanisms, and One Health implications of MDR P. aeruginosa, supported by up-to-date literature and well-organized thematic sections. Figures and tables are informative and enhance the clarity of complex concepts. The discussion successfully integrates microbiological, clinical, and epidemiological perspectives, offering valuable insights for both practitioners and researchers. While the review is detailed and insightful, minor revisions are recommended to streamline some dense sections and to strengthen the applicability of proposed surveillance and control strategies. Overall, the manuscript represents a significant contribution to the field of veterinary infection control and antimicrobial resistance surveillance.
Please see the attached file for the comment

Author Response
Dear Reviewer 3,
We sincerely thank you for your encouraging assessment of our review and for your careful, detailed suggestions. Below, we address every remark in the order it appeared in your annotated PDF.
